# Mechanical Properties and Fracture Behavior of an EBW T2 Copper–45 Steel Joint

**DOI:** 10.3390/ma12101714

**Published:** 2019-05-27

**Authors:** Peng Liu, Jiafeng Bao, Yumei Bao

**Affiliations:** 1Key Laboratory of E&M, Ministry of Education & Zhejiang Province, Zhejiang University of Technology, Hangzhou 310014, China; liu_394962328@163.com (P.L.); baojfjob@sina.com (J.B.); 2Zhijiang College of Zhejiang University of Technology, Shaoxing 312030, China

**Keywords:** dissimilar welding joint, mechanical properties, small punch test, microstructure, fracture behavior

## Abstract

The dissimilar joining of T2 copper to 45 steel was performed by electron beam welding (EBW). Full-strength joints were obtained, and the highest tensile strength was found to be 270 MPa, which is almost equal to the strength of copper. Moreover, the macroscopic morphology of the tensile fracture exhibited an obvious necking phenomenon and features such as dimples, and spherical structures were found via scanning electron microscopy (SEM). These results indicated that the fracture of the T2 copper–45 steel joint is a mixed mode of cleavage and ductile fracture. Meanwhile, the fracture toughness was determined using the small punch test (SPT) with a drop rate of 0.5 mm/min. SEM imaging of the fracture surfaces revealed that the fracture was controlled by microscopic void nucleation and always occurred in the copper-side heat affected zone (HAZ). Finally, mutual verification between the numerical simulation of the finite element and the SPT results confirmed that the fracture first occurred in the copper-side HAZ due to the toughness difference.

## 1. Introduction

With the development of technology, materials with energy-saving lightweight and multifunctional properties are becoming more and more common [1]. However, the performance of a single-metal material is difficult to enhance. In contrast, the connection of dissimilar metals appears to have a good development prospect [2].

Copper alloys have been widely applied in aerospace, microelectronics, and metallurgy fields due to their excellent performances, such as their high thermal conductivity (401 W·m^−1^·k^−1^), high conductivity (≥56 Ms·S^−1^), high melting point (1083 °C), and good ductility [3]. However, the application of copper is limited due to its low intensity and high cost while 45 steel is a common material with a high intensity, a high melting point (1495 °C), a low thermal conductivity (36~54 W·m^−1^·k^−1^), and a low cost in most industry fields [4]. Therefore, a composite structure composed of T2 copper and 45 steel should be able to achieve excellent complementary performances.

The connection of dissimilar metals can achieve superior performances in comparison to the properties of the same metals on their own, but the realization of their connection faces a series of problems. Differences in the physical and chemical properties and the composition of the two metals—including the melting point, linear expansion coefficient, compatibility, thermal conductivity, and specific heat capacity—make defect-free dissimilar welding difficult [5]. For instance, there is a good foundation for the welding of steel and copper because there is no intermetallic compound and the two materials have a similar crystal structure. However, the welding of copper induces a large grain size because of the excess heat input and high thermal conductivity, which worsen the performance of the joint. In this paper, we concentrate on electron beam welding (EBW), the advantages of which—including deep penetration, a small heat affected zone (HAZ), and the cutting off of the adverse effects of air under vacuum [6,7,8]—are presented in connection with the welding of dissimilar metals. There is a large number of articles published on this topic, but most of them only address how to improve the weld quality and the mechanical properties of joints of a certain thickness, while the material properties of miniature thickness joints are less studied [9,10,11]. Thus, the connection of copper and steel using EBW is still of high interest.

At present, there are many methods used to test the mechanical properties of joints. The small punch test (SPT), as a new kind of small specimen technology, was proposed in the early 1980s. Significant research has proved that it can be used to estimate the fracture properties of in-service material with miniature-sized specimens [12]. To extend the application areas of this method, various correlations have been established between fracture parameters obtained from standard tests and the load–displacement (L-D) curves of the small punch test [13,14,15,16]. However, it remains rarely employed in the field of testing dissimilar metal joints.

Obviously, it is not enough to analyze the deformation characteristics of dissimilar joints from limited experiments. An increasing number of researchers have turned to finite element analysis to solve some complicated analyses in SPT [17,18]. Particularly in the discussion of elastic–plastic damage behavior, finite element analysis presents a significant advantage [19]. However, few studies have focused on deformation behavior, which can provide a comprehensive analysis of SPT in order to optimize the approaches to mechanical strength evaluation.

In this study, a combination of experimental and numerical analyses was used to investigate the elastic and plastic deformation behavior of T2 copper–45 steel. Based on a tensile test, the basic parameters and performance of dissimilar welded joints were analyzed. Meanwhile, on the basis of damage fracture theory, a finite element model was established, the results of which are consistent with the SPT conclusions. The microscopic morphology of the fracture was observed via scanning electron microscopy (SEM) and determined the T2 copper–45 steel joint fracture to be a mixed fracture mode of cleavage fracture and ductile fracture.

## 2. Materials and Methods

### 2.1. Preparation of the T2 Copper–45 Steel Joint

The material analyzed in this paper was a T2 copper-45 steel EBW joint, which has been increasingly employed in the field of engineering. The initial dimensions before welding were 220 × 200 × 2.5 mm^3^. The chemical compositions of T2 copper [20] and 45 steel [21] are listed in Table 1; Table 2, respectively. The EBW parameters of the T2 copper–45 steel joint were as follows: Acceleration voltage 60 kV, electron beam 80 mA, vacuum degree 5 × 10^−2^ torr, welding speed 300 mm/min (SEBW welding system, Guilin, China). In order to ensure a good surface feature of the joints, the surface was polished by a grinding wheel (DSD250, Metabo, Gottingen, Germany).

### 2.2. Tensile Test and SPT Procedure

The tensile specimen was obtained by wire cutting, and the conventional mechanical properties of the material were estimated using the standard GB/T228.1-2010 (Metallic Materials—Tensile—Part 1: Method of testing at room temperature) [22]. The specimen size for the tensile tests and sampling orientation (NP), conducted at room temperature, is shown in Figure 1. The ultimate strength, the yield strength, and the Young’s modulus were determined using the INSTRON-8801 Servohydraulic Fatigue Testing System (INSTRON, Shanghai, China) with a loading rate of 1 mm/min. A VEGA 3 SBH scanning electron microscope (TESCAN, Prague, Czech Republic) was used to observe the microstructure and fracture appearance in the joint. The magnification was set to be 1000× in a vacuum environment.

As shown in Figure 2, the self-designed SPT device including a specimen holder, a punch, and a steel ball (2.4 mm in diameter) was fitted on an INSTRON-5869 testing machine (INSTRON, Shanghai, China). Prior to the experiment, the round thin disk was ground, and the final dimension was d 10 × (0.5 ± 0.02) mm. In the experiment, the specimen was first placed in the lower die, after which the upper die and lower die were screwed in. Finally, the punch ball was placed in the hole and aligned with the center of the specimen. The SPT investigation was carried out with a drop rate of 0.5 mm/min. The deformation of the specimen occurred in the center, and the L-D curve was recorded under the indentation of the ball.

### 2.3. Numerical Simulation

To predict the deformation behavior accurately, a sufficiently fine finite element calculation of SPT was necessary to ensure a good representation of the deformation process. In previous research, different models were proposed to analyze the SPT [22,23,24,25]. In this research, the stress–strain field of the weld zone was obtained from a previous welding simulation of the reference method [26] and an established a material model based on shear damage (damage for ductile metals). The density of the grid was controlled by seeding the sides of the specimen, and standard explicit linear 3D stress elements were used. There was a small amount of C3D8R wedge mesh in the center of the circle. The degree of freedom in all directions of the circumference was limited to 0, and the punch ball was simplified into a rigid body, while the factor of mass scaling was set to 100. The schematic of the finite element model in this research is shown in Figure 3. Meanwhile, the parameters of damage evolution for T2 copper–45 steel were defined. For T2 copper, the values of the fracture strain, shear stress ratio, and strain rate were fixed at 0.5, 1.8, and 0.001, respectively. The corresponding values of 45 steel were 0.3, 1.5, and 0.001. The value of *K_s_* can be fixed to 0.03 and 0.01 for copper and steel, respectively.

## 3. Results and Discussion

### 3.1. Analysis of the Tensile Test and SPT

Tensile properties are the most basic mechanical properties for measuring the quality of welded joints [27]. In the tensile test, the L-D curve is the most intuitive reflection of specimen deformation. The L-D curve of the tensile specimen is shown in Figure 4. In the figure, three distinct deformation phases (elastic deformation, plastic deformation, and necking deformation) are displayed. An approximate straight line was found in the elastic zone of 0–0.18 mm, and the maximum load value was around 5.387 kN, located in the plastic phase of 0.18–1.64 mm. The final fracture displacement of the specimen was 2.38–3.01 mm.

As shown in Table 3, the basic properties of the T2 copper–45 steel joint were obtained by the tensile test, and the basic properties of the base metal are listed in Table 4. It is obvious that the joint exhibited good mechanical properties as the yield stress was 3.73 times that of T2 copper, and the tensile strength exceeded the low strength of the T2 copper base metal at 112%. Meanwhile, the Young’s modulus also approached that of 45 steel. The elongation exceeded 5%, which was the minimum value specified for ductile material.

In the SPT, the elastic and plastic phases of the L-D curves were the focus of most concern. Similarly, an approximate straight line could be found, and this region was mainly controlled by the elastic properties of the joint. The other four phases are also visually reflected in Figure 5:Zone I: Elastic bending;Zone II: The transition between elastic and plastic bending;Zone III: Plastic hardening;Zone IV: Softening due to material damage initiation;Zone V: Crack growth with a circular shape around the center of the specimen until failure.

The load–displacement curve obtained from the SP tests allowed us to evaluate standard engineering properties, particularly yield strength, ductility, and fracture energy. Mao and Takahashi [28] suggested some equations for predicting yield strength and tensile strength separately from the measured small punch L-D curve. It was assumed that the material exhibited elastic-power law plastic behavior, and the following correlation was found between the maximum small punch load and the tensile strength, *σ_UTS_*:(1)$$\sigma_{UTS}=130\left(\frac{P_{max}}{t_{0}^{2}}\right)-320$$

Mao and Takahashi also defined a load on the small punch curve, *P_y_*, as the load where initial non-linearity is observed on the small punch L-D curve depicted in Figure 5. Another relationship was suggested that correlated *P_y_* with the material yield strength as in Equation (2). The joint was recognized as a ductile material (materials with an elongation exceed 5% are considered to be ductile materials in the field of engineering), in which fracture toughness is an important property. The fracture toughness of EBW T2 copper–45 steel can be estimated from the L-D curve as follows [29]:(2)$$\sigma_{y}=360\left(\frac{P_{y}}{t_{0}^{2}}\right)$$
(3)JIC=42(δ*t0)3/2−50
where *σ_y_* is the yield strength, *t*_0_ is the initial thickness, and *δ*^*^ is the fracture deflection of the small punch specimen.

As shown in Figure 6, elastic deformation played a key role, and the void volume fraction remained unchanged. Then, a transition was observed from elastic deformation to plastic deformation, and the macroscopic deformation of the small punch specimen gradually became obvious as the void began to nucleate along with crack initiation. The final fracture displacement of the small punch specimen was around 1.68–1.82 mm.

After the aforementioned judgement, these parameters were calculated using Equations (1)–(3), which are presented in Table 5.

### 3.2. Analysis of Fracture Behavior

The fracture location of the T2 copper/45 steel joint is shown in Figure 7. The fracture zone exhibited significant necking, which revealed apparent ductile fracture characteristics. The fracture mechanism was mainly manifested in the difference of thermal conductivity between T2 copper and 45 steel. Because the heat distribution during the welding process was mainly concentrated on the copper side, this led to the formation of coarse grains on the copper side and resulted in the copper-side HAZ bringing the weakest zone of the T2 copper–45 steel joint. For the weld zone, the cooling rapidity characteristic of EBW generated fine grains in the weld zone, which resulted in a relatively higher weld strength. It can be seen from Figure 7c that the copper-side HAZ underwent obvious plastic deformation while the steel-side HAZ was the most absent. This is because the fracture in the copper-side HAZ could be attributed to a large amount of Cu species diffusing into Fe rather than the reverse, leading to the possible formation of microvoids in Cu, while the steel side remained uninfluenced [10]. These results closely matched the findings of previous research [10,11]. Furthermore, the strength of the steel-side HAZ was found to be significantly greater than the copper-side HAZ.

It can be seen from Figure 8a that the morphology of the tensile fracture showed spherical structures and microvoids, which were associated with the microscopic features of crystal brittleness and cleavage fracture. The fracture surface shown in Figure 8b indicated a large number of shallow dimples. Moreover, the shape of the dimples was equiaxed, suggesting that the joint had a certain plastic deformation ability. The morphology depicted in Figure 8c showed shallow dimples, cleavage facets, ledges, and terraces, which are all associated with cleavage fracture. The visible defects (the unmelted region) and gas pores, shown in Figure 8c, caused stress concentration, thus inducing crack growth. Therefore, it was concluded that the fracture types in the specimen were cleavage fracture and ductile fracture in a mixed model. There were no significant differences between the fracture surfaces of the different specimens.

As shown in Figure 9, the plastic deformation was mainly reflected in the annular Erichsen shape of the disk [30]. As the punch ball continued downward and the plastic deformation ended, a microcrack first appeared in the copper-side HAZ on both sides. With the increase of the central deflection of the annular zone in the specimen, the microcrack and damage would gradually form and extend. When the external load was further continued, the main crack expanded rapidly and caused obvious macroscopic cracks on the left copper-side HAZ. The secondary crack caused on the right copper-side HAZ was the result of the further expansion of the microcrack. Meanwhile, the obvious increase of outgrowth was also found, indicating the accumulation of damage. The outer reaches of the initiation region (Figure 9aA) had a large amount of slip-band [30], and the deformation was more gathered in the copper side, which revealed that the accumulation of damage mainly existed in the initiation region but the fracture first occurred in the initial crack region (Figure 9: Point c).

As shown in Figure 10, dimples and spherical structures appeared in the fracture surface of the SPT specimen, indicating cleavage fracture and ductile fracture, respectively. The mechanism involved microvoid nucleation (point a to b), cavity growth, the aggregation of adjacent cavities to form cracks (point b to c), and crack propagation resulting in the fracture (point c). The main source of the void nucleation was the difference in the physical properties of T2 copper and 45 steel. In particular, the thermal conductivity of T2 copper is much higher than that of 45 steel, resulting in a large amount of heat being biased to the copper side during the welding process. Thus, the copper-side growth increased seriously, and the grain size was coarse. The ductile–brittle transition temperature of the bi-material increased, resulting in a decrease of toughness. Consequently, this region became the weakest region. Meanwhile, coarse grain regions and a large void (defect) size both induced the formation of microscopic voids. 

Regardless of the reason for their formation, the microscopic voids continued to grow under the external forces, and the adjacent cavities connected with each other to form cracks. With the continuous external forces, the cross-section of the matrix between the cavities continued to shrink. Eventually, these circumstances led to fracture.

### 3.3. SPT Simulation

The crack initiation position of the sample in the finite element simulation results is shown in Figure 11. It can be clearly seen that the results were similar to the experimental results. The fracture of the sample did not completely occur at the center of the sample, and a linear crack occurred after the fracture displacement was reached. In the finite element analysis results, the variable output was output frame by frame, so the fracture time of the sample could not be accurately obtained in the finite element result. However, it could be determined that the fracture occurred in the HAZ near the copper side. In addition, since the crack could be considered to be caused by cell deletion, the width of the crack was found to be related to the finite element mesh size. It can be seen that in the range of 0.0125 s after the start of the crack, the crack propagation speed was extremely fast, and a large-sized macrocrack was instantaneously formed.

As can be seen in Figure 11, the stress was mainly concentrated in the center of the circle, and the steel-side stress was significantly larger than the copper-side stress because the toughness of copper is obviously better than the toughness of steel. As the ball center shifted toward the steel side with the depression of the small ball, the stress was mainly concentrated on the steel side. The stress dropped from the central circle to the circumference. The fracture first appeared in the copper-side HAZ and had a tendency to expand toward the steel side. Furthermore, it is obvious from Figure 12 that the experiment and simulation were consistent in the increase of displacement.

## 4. Conclusions

(1)Based on the tensile test, the ultimate strength of the T2 copper–45 steel joint was determined to be 267.54 MPa, the yield strength was 240.93 MPa, and the Young’s modulus was 174.28 GPa. The fracture toughness was then determined to be 210.827 KJ·m^−2^ using SPT.(2)The tensile test showed that the fracture was located in the copper-side HAZ because the thermal conductivity of copper is much higher than that of steel, which implied a large amount of heat being biased on the copper side during the welding process. Therefore, the grain became too coarse and resulted in the copper-side HAZ being the weakest joint region. With the future development of deformation, the specimen ended with apparent necking.(3)SPT at room temperature showed that the cracks were first generated by microvoid nucleation and cavity growth. While a macrocrack was formed, a secondary crack also appeared on the opposite side. After SPT was completed, the sign of the Erichsen shape and the outgrowth could be seen on the surface of the specimen intuitively. The appearance of slip-band indicated the offset between the initial accumulation of damage and the initial crack region. Moreover, the fracture types of the specimen were found to be cleavage fracture and ductile fracture, as determined via SEM. Based on theoretical and ABAQUS analyses, it was concluded that the crack first appeared in the copper-side HAZ, and the deflection was controlled by the toughness difference.

## Figures and Tables

**Figure 1 materials-12-01714-f001:**
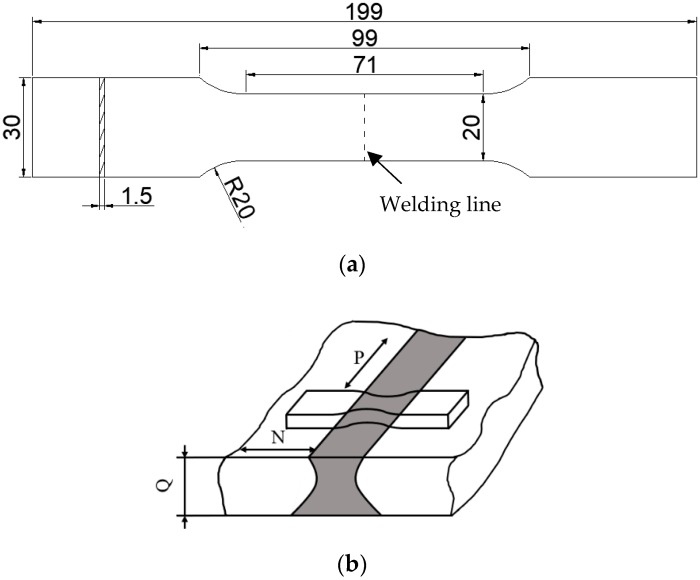
The tensile test specimen: (**a**) the geometry and size of the tensile specimen (unit: mm), (**b**) sampling orientation.

**Figure 2 materials-12-01714-f002:**
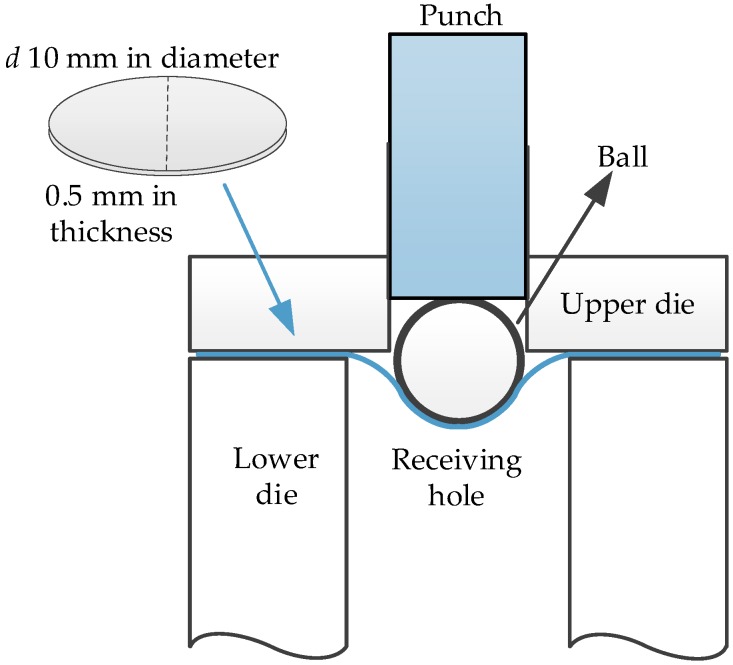
The schematic diagram of the small punch test (SPT).

**Figure 3 materials-12-01714-f003:**
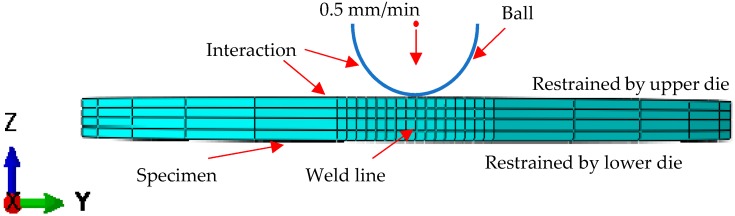
The finite element model of the small punch specimen.

**Figure 4 materials-12-01714-f004:**
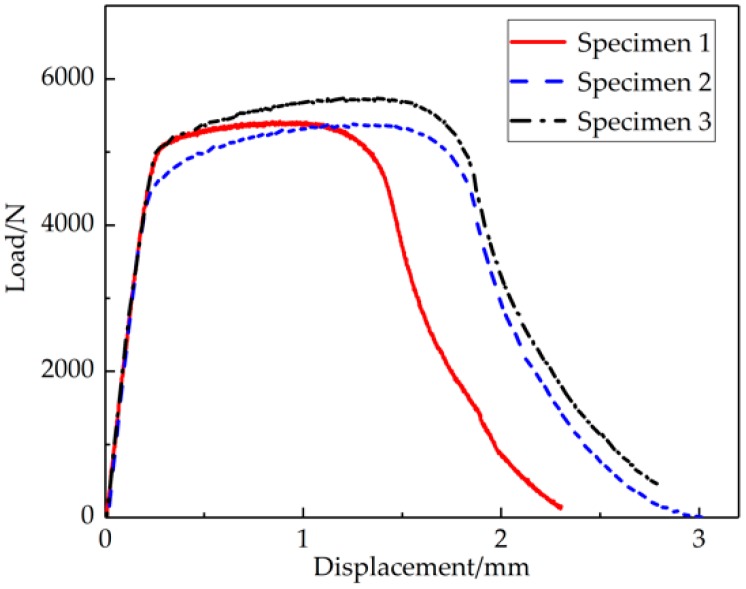
The load–displacement (L-D) curve of the tensile test of the three joints.

**Figure 5 materials-12-01714-f005:**
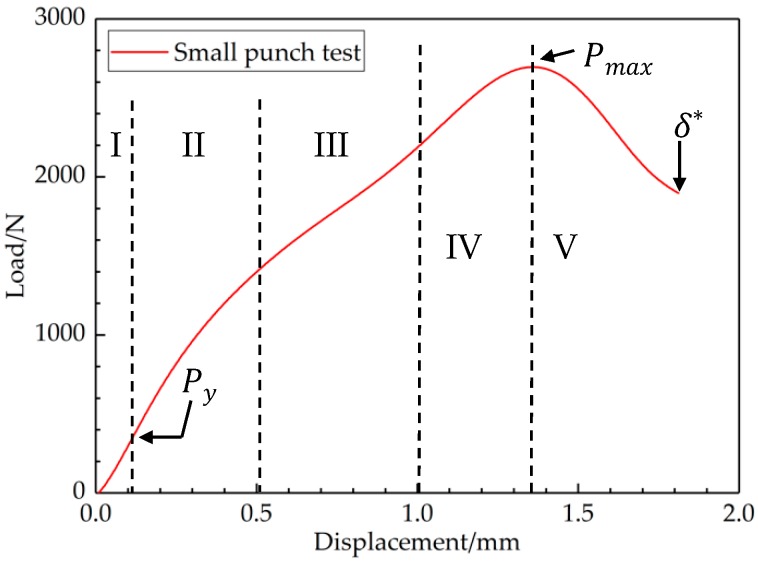
The main behavior zones in the SPT curve.

**Figure 6 materials-12-01714-f006:**
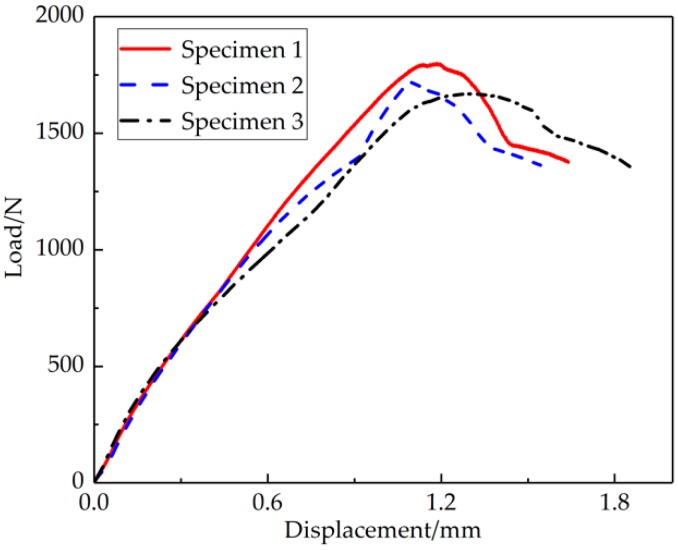
The L-D curve of the SPT of the three joints.

**Figure 7 materials-12-01714-f007:**
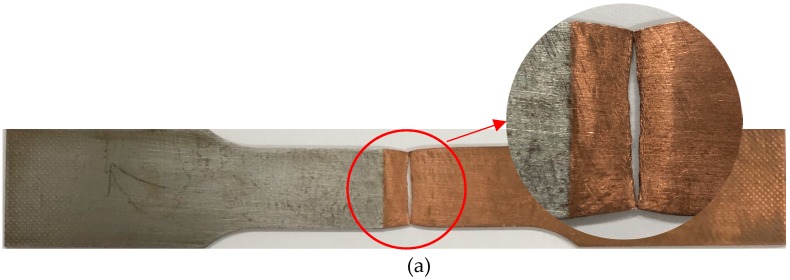
The macroscopic fracture diagrams of the tensile specimens: (**a**) specimen 1, (**b**) specimen 2, (**c**) specimen 3.

**Figure 8 materials-12-01714-f008:**
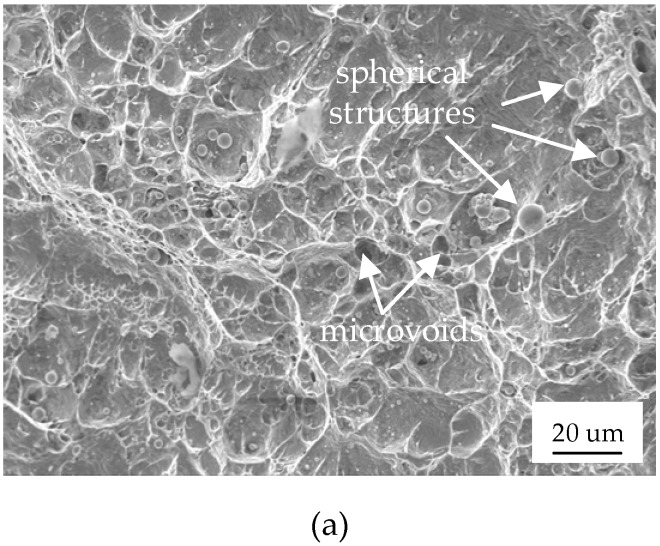
The SEM images of the micro-morphology fracture surface of the tensile specimens: (**a**–**c**) specimens 1, 2, and 3, respectively.

**Figure 9 materials-12-01714-f009:**
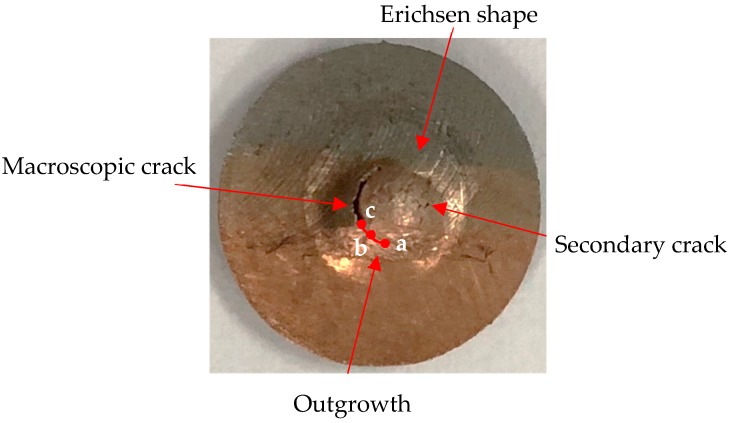
The macromorphology fracture of the SPT specimen: (**a**) the initiation region, (**b**) the middle area, (**c**) the initial crack region.

**Figure 10 materials-12-01714-f010:**
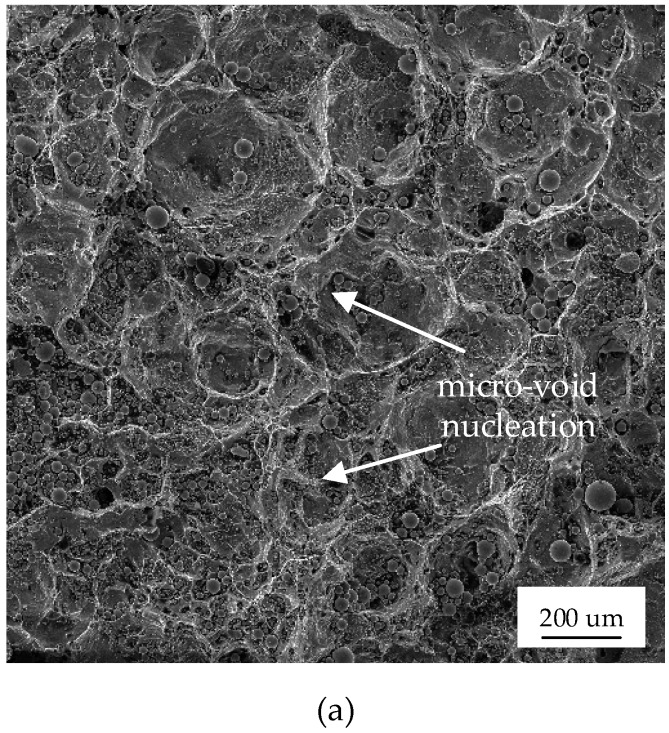
The SEM micro-morphology fracture surface of the SPT specimen: (**a**–**c**) in Figure 9 correspond to points (**a**–**c**), respectively.

**Figure 11 materials-12-01714-f011:**
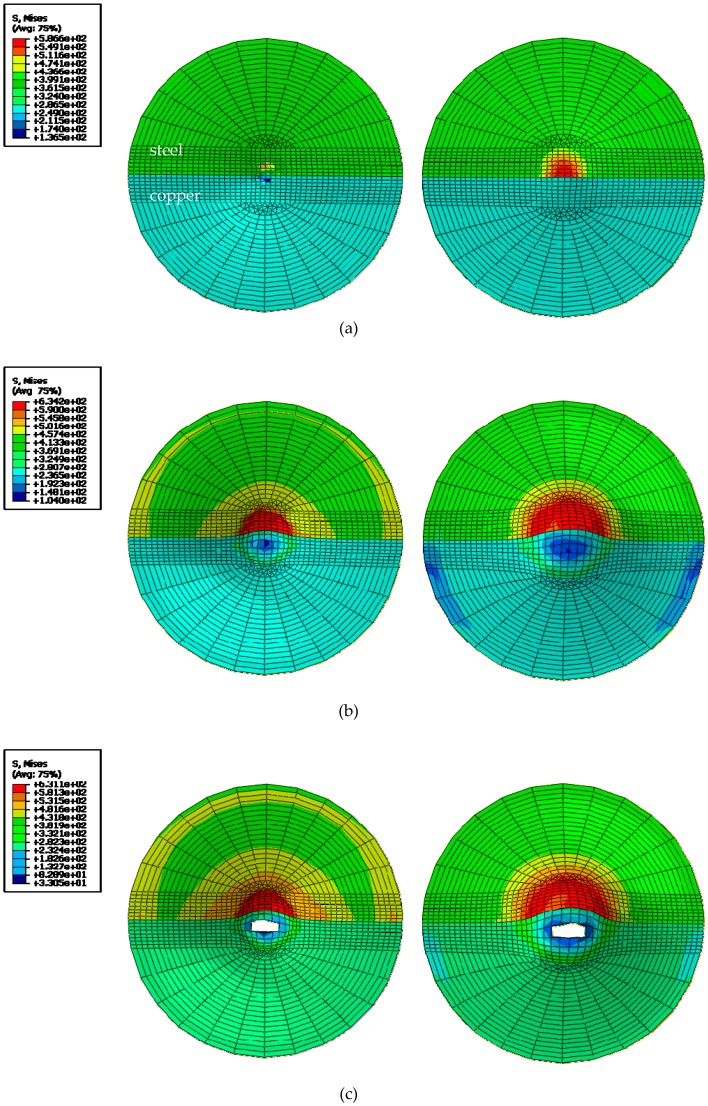
The SPT fracture simulation: (**a**), the initial stage, (**b**) the middle stage, (**c**) the initial cracking stage.

**Figure 12 materials-12-01714-f012:**
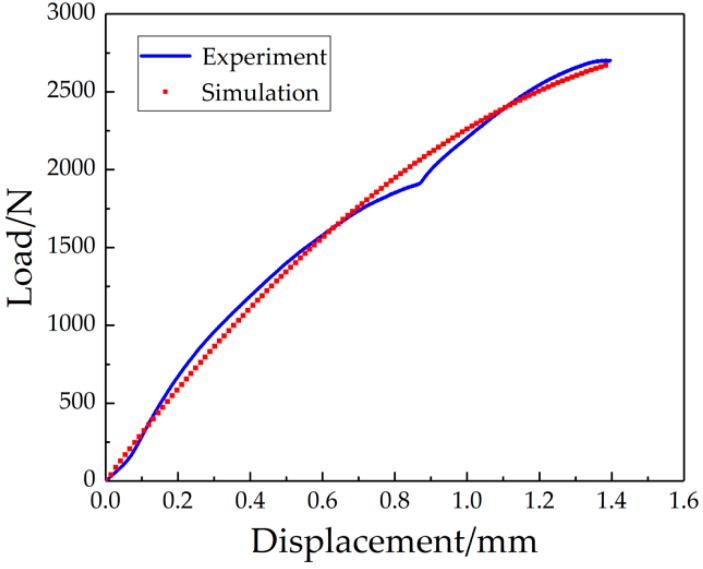
An experiment and simulation comparison of the L-D curve.

**Table 1 materials-12-01714-t001:** The chemical composition of T2 copper (wt %).

Material	Cu + Ag	Pb	Fe	Sb	S	As	Bi	Other
T2 copper	99.9	0.005	0.005	0.002	0.005	0.002	0.001	0.1

**Table 2 materials-12-01714-t002:** The chemical composition of 45 steel (wt %).

Material	C	Si	Mn	P	S	Cr	Ni	Cu
45 steel	0.42–0.50	0.17–0.37	0.50–0.80	0.035	0.035	0.25	0.25	0.25

**Table 3 materials-12-01714-t003:** The tensile test results.

No.	Yield Stress (MPa)	Ultimate Tensile Stress (MPa)	Young’s Modulus (GPa)
1	229.03	262.51	169.24
2	257.24	270.76	178.69
3	236.51	269.34	174.90
Average	240.93	267.54	174.28

**Table 4 materials-12-01714-t004:** The mechanical parameter comparison between the joint and the base metal.

Material	Yield Stress (MPa)	Ultimate Tensile Stress (MPa)	Elongation (%)	Young’s Modulus (GPa)
T2 copper	64.58	238.74	50	115.79
45 steel	>350	>600	>16	>200
Electron beam welding joint	240.93	267.54	6	174.28

**Table 5 materials-12-01714-t005:** The *J_IC_* data from our calculations.

No.	*δ*^*^/mm	*P_y_*/kN	*P_max_*/kN	σ*_y_*/MPa	σ*_UTS_*/MPa	*J_IC_*/KJ·m^−2^
1	1.639	0.149	1.797	214.56	614.44	199.266
2	1.566	0.145	1.720	208.8	574.4	182.889
3	1.856	0.153	1.671	220.32	548.4	250.325

$J_{IC}=\frac{J_{IC1}+J_{IC2}+J_{IC3}}{3}=210.827\;\mathrm{KJ}\cdot m^{-2}$

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
