# Peer review of "Mechanical Properties and Fracture Behavior of an EBW T2 Copper–45 Steel Joint"

_materials, 2019, doi:10.3390/ma12101714_

Round 1
Reviewer 1 Report
The authors’ present a paper concerning the mechanical properties and fracture behavior of electron beam welded Cu-Fe bimetals. The paper is well structured, however there are different unadressed issues.
The material models used for the finite element analysis should be given. It is not said if the mechanical properties in table 4 were measured or taken from literature. If the latter is the case, a source has to be given.
It is not clear if the specimen in figure 6 show the results of the welded specimen or the base material. If the welded material is investigated, it should be discussed in detail if the used formulas are applicable. Generally, the caption of each figure should be more precise, i.e. write which material or material combination is investigated.
How was the specimen positioned in the SPT? The error resulting from the welded zone being not exactly in the middle should be discussed and quantified.
Furthermore, the numerical method could be explained more precisely, e.g. grid size, clamping forces, mass scaling, stiffness of active elements.
The same colorbars with a larger font should be used in figure 11 for all results. A comparison between the calculated stress values and the FEM results should be given.
Why is the crack in figure 9 on a different position as the one calculated by FEM?
I don’t think that the elongation bigger elongation of the copper is responsible for the absence of the plastic deformation of the steel. Only the higher strength should be responsible for that.
In summary, the content of the paper does not seem to be very innovative. It remains an open question what the big achievement of this publication is, as the material combination steel/copper has been investigated by many authors. All in all, I propose to substantially revise the paper.
Author Response
The response was attached below (Cover Letter-1).

Reviewer 2 Report
The article prestents testing of mechanical properties and fracture
behavior of EBW T2 copper – 45 steel joint. The dissimilar joining of T2
copper to 45 steel was performed by electron beam welding.
It was hard to review, because the number of lines have been deleted.
My suggestions/questions:
Section: "1. Introduction"
- please add to your manuscript some info about testing of copper-steel joint presented in the literature. There are a lot of articles in this field, so you should mark the novelty of your work.
Section: 2. Materials and methods"
- Table 1. and Table 2. - You have presented the chemical properties od base materials. Please add info about the source of these values (eg standard, analysis, producer data...).
- please add info about melting temperature of used materials and their properties that names you have listed in Introduction.
- you have to add dimensions of used materials and prepared joint(s) - in the title you have written about joint, have you prepared only one joint?
- please add info about standard describes the rules of tests that you have prepared.
- If you can, please add photo of used EBW stand.
Section: "Discussion3. Results and Discussion"
- there isn't any info from which places from the joint the samples for testing have been taken? Please add schema in the previous section.
- Figure 7 - you have presented picture a), b) and c). Howeere there isn't any description in the name of the figure, what these letters mean. Please add. The same situation with Figure 8.
- paragraph between Figure 9 and Figure 10 - "In particular, the thermal conductivity of T2 copper is much higher than 45 steel, resulting in a large amount of heat being biased to the copper-side during the welding process." - please add values of therman conductivity.
- Figure 11 - the same situation like in Figure 7 and 8. You have to add info what a), b) and c) mean.
Author Response
The response was attached below(Cover Letter-2).

Round 2
Reviewer 2 Report
The article prestents testing of mechanical properties and fracture behavior of EBW T2 copper – 45 steel joint. The dissimilar joining of T2 copper to 45 steel was performed by electron beam welding.
Authors
have replied properly to the most of questions and comments that I
pointed out in the first round of review. Efforts of authors are
appreciated. The quality of this paper has improved a
lot based on the authors' revision.
In my opinion it can be published in present form.